# COVID-19 Vaccination in Pregnancy: Pilot Study of Plasma MicroRNAs Associated with Inflammatory Cytokines after COVID-19 mRNA Vaccination

**DOI:** 10.3390/vaccines12060658

**Published:** 2024-06-14

**Authors:** Ching-Ju Shen, Yen-Pin Lin, Wei-Chun Chen, Mei-Hsiu Cheng, Jun-Jie Hong, Shu-Yu Hu, Ching-Fen Shen, Chao-Min Cheng

**Affiliations:** 1Department of Obstetrics and Gynecology, Kaohsiung Medical University Hospital, Kaohsiung Medical University, Kaohsiung 807, Taiwan; chenmed.tw@yahoo.com.tw; 2Institute of Biomedical Engineering, National Tsing Hua University, Hsinchu 300, Taiwan; peggy1240309@gmail.com (Y.-P.L.); lionsmanic@gmail.com (W.-C.C.); chloe.natshuun@gmail.com (S.-Y.H.); 3Division of Gynecologic Oncology, Department of Obstetrics and Gynecology, Chang Gung Memorial Hospital at Linkou, College of Medicine, Chang Gung University, Taoyuan 333, Taiwan; 4Department of Obstetrics and Gynecology, New Taipei City Municipal Tucheng Hospital, New Taipei City 236, Taiwan; 5International Intercollegiate Ph.D. Program, National Tsing Hua University, Hsinchu 300, Taiwan; 6School of Traditional Chinese Medicine, Chang Gung University, Taoyuan 333, Taiwan; 7Taiwan Business Development Department, Inti Taiwan, Inc., Hsinchu 302, Taiwan; michelle@intilabs.com (M.-H.C.); winterhong@intilabs.com (J.-J.H.); 8Department of Pediatrics, National Cheng Kung University Hospital, College of Medicine, National Cheng Kung University, Tainan 701, Taiwan

**Keywords:** COVID-19 vaccine, mRNA vaccine, microRNA, S1RBD IgG, pregnancy, cytokine

## Abstract

Background: The impact of mRNA COVID-19 vaccines on the immunological profiles of pregnant women remains a crucial area of study. This research aims to explore the specific immunological changes triggered by these vaccines in this demographic. Methods: In a focused investigation, we examined the effects of mRNA COVID-19 vaccination on microRNA expression in pregnant women. Key microRNAs, including miR-451a, miR-23a-3p, and miR-21-5p, were analyzed for expression changes post-vaccination. Additionally, we assessed variations in S1RBD IgG levels and specific cytokines to gauge the broader immunological response. Results: Post-vaccination, significant expression shifts in the targeted microRNAs were observed. Alongside these changes, we noted alterations in S1RBD IgG and various cytokines, indicating an adapted inflammatory response. Notably, these immunological markers displayed no direct correlation with S1RBD IgG concentrations, suggesting a complex interaction between the vaccine and the immune system in pregnant women. Conclusions: Our pilot study provides valuable insights into the nuanced effects of the mRNA COVID-19 vaccine on immune dynamics in pregnant women, particularly emphasizing the role of microRNAs. The findings illuminate the intricate interplay between vaccines, microRNAs, and immune responses, enhancing our understanding of these relationships in the context of pregnancy. This research contributes significantly to the growing body of knowledge regarding mRNA COVID-19 vaccines and their specific impact on maternal immunology, offering a foundation for further studies in this vital area.

## 1. Introduction

In November 2019, there was an outbreak of pneumonia of unknown cause in Wuhan, China, and a new coronavirus “SARS-CoV-2” emerged [1]. In just over two months, the number of confirmed cases of “COVID-19” worldwide exceeded 10,000. This is a rapid spread of disease unprecedented in this century. As of May of 2024, the epidemic continues to undergo changes [2]. Although scientists have made efforts to accelerate research, the characteristics and trajectory of the epidemic remain unpredictable and rife with variables. To ameliorate the disastrous effects brought on by COVID-19, vaccination is the most effective response. In response to the urgency of the epidemic and the needs of the world, the COVID-19 vaccine was developed and released to the public following rigorous efficacy and safety testing [3,4]. Effective vaccination strategies or approaches being necessary to arrest the pandemic, some studies have noted that mRNA vaccines such as the Moderna (mRNA-1273) and Pfizer-BioNTech BNT162b2 vaccines appear to be more effective among all the types of COVID-19 vaccines currently available [5,6,7,8,9].

In special groups, vaccination strategy must be more carefully considered. Vaccination is completely safe during pregnancy and can provide excellent protection for newborns [10,11,12]. Both the fetus and pregnant women are at risk of infection during pregnancy [13,14]. The American College of Obstetricians and Gynecologists (ACOG) and the Society for Maternal–Fetal Medicine (SMFM) recommend that pregnant or breastfeeding women get the COVID-19 vaccine [15]. From 2019 to now, multiple studies have confirmed that administering the COVID-19 vaccine during pregnancy does not lead to an increased risk of adverse perinatal outcomes [16,17].

MicroRNAs can regulate gene expression in many organisms, affect the mechanism of human cancer, and are highly valued molecules in modern medicine [18]. After binding to the three-terminal untranslated region (3’UTR) [19] of the target mRNA, the regulation of gene expression ensues, and protein translation is promoted or inhibited. There is some evidence that some microRNAs are dysregulated in COVID-19 patients. One study identified miR-146a-5p, miR-21-5p, and miR-142-3p as the potential biomarkers of COVID-19 severity that could also be therapeutically targeted [20]. Another study indicated that miR-423-5p, miR-23a-3p, and miR-195-5p could independently identify COVID-19 cases and distinguish SARS-CoV-2 from influenza infection [21]. Many studies examine microRNAs in patients with COVID-19, but little research is available on the regulation of microRNAs following COVID-19 vaccination. Yusuke Miyashita et al. [22] identified several immunomodulatory microRNAs, including miR-126, miR-132, miR-221, and miR-625-3p, that were associated with partial immune responses to COVID-19 vaccination. Additionally, miR-126 and miR-451a correlate with TNF-α levels after receiving the initial dose of the COVID-19 vaccine.

There are also many immune-related cytokines associated with the regulation of microRNAs. For example, miR-451a is an IL-6R translational repressor that exacerbates the IL-6-induced cytokine storm [23] and one of the top five downregulated microRNA genes in COVID-19 patients [24]. One study demonstrated that the presence of miR-451a significantly suppressed TNF-α and significantly increased IL-10 [25]. Another scientific study showed that miR-21-5p had an anti-inflammatory effect that could inhibit the expression of IL-1β [26,27]. To better understand immune response following the administration of mRNA vaccine and the relationship between related cytokines and microRNA, we conducted a series of studies on 111 pregnant women. Figure 1 shows the experimental design of our pilot study.

## 2. Materials and Methods

### 2.1. Participants and Sample Collection

This prospective study was approved by the Institutional Review Board of Kaohsiung Medical University Hospital (IRB No. KMUHIRB-SV(II)-20210087). The study enrolled pregnant women aged 20 or above who had no history of preterm labor, chronic illnesses requiring immunosuppressants, cancer requiring specific treatments, pregnancy-related illnesses such as gestational hypertension or diabetes, or previous COVID-19 disease and SARS-CoV-2 infection. All the eligible participants provided written informed consent before enrollment. We specifically selected the cases where the interval between the last dose of the vaccine and delivery was between 4 and 8 weeks because the antibody concentration generated during this period was at its peak (plateau phase). This time window ensured the highest antibody transfer efficiency to the fetus as well. In addition, we collected and analyzed liver and kidney function tests and coagulation profiles for the participants who underwent cesarean section. The coagulation function test was performed for the participants who received painless labor during vaginal delivery. These biochemical parameters were within normal ranges, meaning no significant adverse effects on the liver, kidney, or coagulation function in our study population.

Peripheral blood samples were obtained for microRNA analysis from 13 mothers on the day of delivery and plasma samples were obtained from 3 non-pregnant healthy women. Additionally, peripheral blood samples were collected for ELISA analysis from 111 mothers on the day of delivery. Regarding the 111 patients from whom the ELISA analysis samples were obtained, 54 completed three doses of the vaccine, 49 completed two doses of the vaccine, and 8 completed one dose. Both the two-dose vaccine and the one-dose vaccine recipients received the Moderna (mRNA-1273) vaccine. Among the three-dose recipients, all received the Moderna vaccine as their third dose, whereas their first and second doses were sets of either the Moderna, the Oxford/AstraZeneca, or the Pfizer-BioNTech BNT162b2 vaccine. None of the pregnant demonstrated any symptoms related to COVID-19 during pregnancy.

Combining all the vaccines received, the participants were classified into one of three groups as follows: (1) the three-dose group, whose participants received either three doses of COVID-19 vaccine with Tdap and flu vaccination (n = 23), three doses of COVID-19 vaccine with Tdap vaccination (n = 24), or only three doses of COVID-19 vaccine (n = 7); (2) the two-dose group, whose participants received either two doses of COVID-19 vaccine with Tdap and flu vaccination (n = 7), two doses of COVID-19 vaccine with Tdap vaccination (n = 21), or only two doses of COVID-19 vaccine (n = 21); and (3) the one-dose group, whose participants received either one dose of COVID-19 vaccine with Tdap vaccination (n = 2), or only one dose of COVID-19 vaccine (n = 6).

### 2.2. microRNA qPCR Analysis

Total RNA was isolated from 200 μL of plasma with a miRNeasy Serum/Plasma Advanced Kit (Cat. No. 217204, Qiagen, Hilden, Germany) following the manufacturer’s protocol. The plasma RNA samples were eluted in 20 μL nuclease-free water. The concentration of the extracted total RNA sample was quantified using a Thermo Fisher’s Qubit microRNA Assay Kit (Q32880, Hsinchu, Taiwan).

We investigated the real-time microRNA expression profiles with a MIRAscan and NextAmp™ Analysis System (designed by Inti Taiwan, Inc.; manufactured by Quark Biosciences, Inc., Zhubei, Taiwan), which was pre-printed with microRNA-specific primers. To identify differentially expressed microRNAs, the microRNA expression profiles were first normalized by using the quantile normalization method. The resulting microRNA expression profiles were normalized by the expression level of the qPCR spike-in control using the following formula:
∆Cq value=20−(Cq value−control)

The higher the resulting value, the higher the microRNA expression level. Any microRNAs without amplification signals across all the profiles were removed. The differentially expressed microRNAs were filtered by 2 criteria: delta Cq ≥ 0.585 or ≤−0.585 and *p*-value < 0.05.

For microRNA enrichment analysis, we used the differentially expressed microRNAs as input and analyzed microRNA target interaction (MTI) in miRTarBase [28]. miRTarBase is one of the largest experimentally validated databases for microRNA target analysis. MTIs with more than 1 strong evidence or more than 2 weaker evidence or paper reports were retained. Finally, the filtered target gene list was used for the enrichment analysis using the R package clusterprofiler [29] according to the pathways or gene functional groups from Gene Ontology (GO), KEGG Pathway, Reactome pathway, WikiPathways, Disease Ontology, and DisGeNET [30,31,32,33,34].

### 2.3. ELISA

The concentration of IL-6, IL-6R, TNF-α, IL-10, IL-1β, and human IgG antibody against SARS-CoV-2 S1 RBD protein in plasma were analyzed from the samples taken on the day of delivery using the R & D system Quantikine Human Immunoassay for IL-6, IL-6R, TNF-α, IL-10, IL-1β, and RayBio^®^ COVID-19 S1 RBD protein Human IgG ELISA Kit (RayBiotech, Peachtree Corners, GA, USA).

The different variants of neutralizing antibodies, including the BA.1, BA.2, BA.4, and BA.5 variants, were determined using the GenScript SARS-CoV-2 Surrogate Virus Neutralization Test Kit for wild-type and AcroBiosystem Neutralizing Antibody Titer Serologic Assay Kit. The results were used to calculate the neutralizing antibody inhibition rate based on the following formula:
$$\mathrm{Inhibition}\left(\%\right)=\left(1-\frac{\mathrm{OD}\;\mathrm{value}\;\mathrm{in}\;450\;\mathrm{nm}\;\mathrm{of}\;\mathrm{sample}}{\mathrm{average}\;\mathrm{OD}\;\mathrm{value}\;\mathrm{in}\;450\;\mathrm{nm}\;\mathrm{of}\;\mathrm{negative}\;\mathrm{control}}\right)\times 100\%$$

The experiment was stopped by adding a stop solution and the color change from blue to yellow was evaluated. The OD value absorbance (color intensity) at 450 nm was read using a microplate spectrophotometer (Molecular Devices, San Jose, CA, USA). All the ELISA results were obtained per the manufacturer’s protocol.

### 2.4. Statistical Analysis

The data were analyzed using GraphPad Prism 8 (GraphPad Software, San Diego, CA, USA) and R Project for Statistical Computing (R version 4.2.2, Auckland, New Zealand). The *p*-value was analyzed by using the two sample Student’s *t*-test to compare the two different vaccine combinations, i.e., one, two, or three doses of COVID-19 vaccine with or without Tdap or influenza vaccine. The error bars represent the standard deviation (SD). All the results with *p* < 0.05 were considered statistically significant. Principal component analysis (PCA) was used to evaluate any differences among the samples and treatment conditions. An orthogonal transformation was performed to convert a set of observations of possibly correlated variables into a set of values of uncorrelated variables called principal components.

## 3. Results

### 3.1. Participant Characteristics

Table 1 and Table 2 display the participant demographic and clinical characteristics. Table 1 displays the data for the 16 participants whose plasma samples were used for the microRNA analysis. Table 2 displays the data for the 111 participants whose samples were used for the ELISA analysis. These 111 participants were all vaccinated with the Moderna vaccine as their third dose. The participants who received only one dose or two doses all received Moderna vaccines. For the group of participants who received three doses, the first and second doses of the vaccine were sets of either the Moderna, Oxford/AstraZeneca, or Pfizer-BioNTech BNT162b2 vaccine. No participants demonstrated symptoms related to COVID-19 during pregnancy. The median maternal age in the one-dose group was 34.25 years (IRQ 38–28.5). The median maternal age in the two-dose group was 32.37 years (IRQ 35–28). The median maternal age in the three-dose group was 33.57 years (IRQ 37.75–30). Among the 111 participants, 47 received pertussis vaccine, 30 received influenza and pertussis vaccine, and 34 received neither influenza nor pertussis.

### 3.2. The Different Expression Levels of microRNA

After the microRNA array analysis, we used the following formula to normalize the generated microRNA expression profile through the expression level of the qPCR spike-in control:
∆Cq value=20−(Cq value−control)

The higher the result value, the higher the expression level of microRNA; Figure 2 displays the microRNA values with differential expression from 16 samples. We used principal component analysis (PCA), performing an orthogonal transformation to convert the observed values of a set of possibly correlated variables into the values of a set of uncorrelated variables. The difference is shown in Figure 3. The sample values that deviated significantly from the values obtained from the samples of the subjects receiving the same vaccine dose were excluded. After excluding the discrete samples, the eight microRNAs shown at the bottom of Figure 2 were clearly distinguished during the grouping process. These represented the microRNAs that are expressed in most samples and were used to create a heatmap drawing with a total of eight microRNAs (Figure 4). Three microRNAs exhibited the highest levels of expression, i.e., miR-451a, miR-23a-3p, and miR-21-5p.

We inputted the three microRNAs that showed significant differences in expression levels and then performed a gene set enrichment analysis on the pathways and gene ontology. Among the top five biological process enrichment GO terms shown in Table 3, Table 4 and Table 5, several of them were found to be associated with inflammation or interleukin, suggesting the potential involvement of these microRNAs in regulating inflammatory processes.

### 3.3. The Level of IL-6, IL-6R, TNF-α, IL-10, IL-1β, and S1RBD IgG in Samples from Patients Receiving Different Doses of COVID-19 Vaccine

We carried out two classification methods for the experimental data regarding the quantification of IL-6, IL-6R, TNF-α, IL-10, IL-1β, and S1RBD IgG: (1) classification according to the number of doses administered and (2) classification based on the number of weeks between the third-dose vaccine administration and the time of delivery. Figure 5 displays the trend in cytokine and IgG concentrations. The concentration of both in the samples from the groups receiving three doses of mRNA vaccines was higher than those who received two doses or one dose, and there were also significant differences between the groups. The three-dose group was used for classification. The concentration of IgG antibodies has a downward trend with the number of weeks, and the changes in other cytokines are quite different. From the perspective of IL-6, IL-6R, and IL-1β, the concentration at 7–8 weeks is the lowest. This may be because the time interval from vaccination was longer. Notably, the same results exist for all the cytokines, with the highest concentration at 5–6 weeks. TNF-α appears to be the primary inflammatory factor compared with the other factors. The number of weeks does not seem to have a significant impact on concentration. The concentration of IL-10, a cytokine of anti-inflammatory response, was higher at 7–8 weeks than at 3–4 weeks.

We examined whether the administration of pertussis and influenza vaccination produced changes in cytokine concentration. Figure 6 displays the concentration of each cytokine for the two- and three-dose groups. The cytokine concentration for the samples from the one-dose group is presented in Appendix A. The administration of the pertussis vaccine, pertussis plus influenza vaccine, and the type of vaccine used for the first and second doses (AZ or BNT) do not appear to significantly impact the cytokine concentration. The statistical analysis shows no significant difference (*p* = ns). While the initial comparison was limited to those groups receiving one, two, or three vaccine doses, there were also variations observed within these groups, depending on whether pertussis was present or not. These differences were observed to varying degrees. These findings indicate that we can exclude the pertussis vaccine and influenza vaccine variables. Additionally, the type of first and second doses appears to have no effect on the need for a booster dose of the Moderna vaccine.

To determine whether there was any correlation between the five cytokines and whether they influence each other, we used the R program software (version 4.2.2) to conduct a correlation analysis among the five cytokines. Figure 7 displays the correlation between all the cytokines. Except in the cases of SARS-CoV-2, which creates a cytokine storm, the immune system provides an appropriate response, mediates resistance to invading microorganisms, and enables host survival after infection in general. As the body responds to vaccines, the persistently circulating low levels of pro-inflammatory cytokines may be a natural consequence of innate and adaptive responses to the vaccination, as compared to those diagnosed with COVID-19. After the third or even fourth dose of the COVID-19 vaccine, the body maintains a low level of inflammatory cytokines. We examined the correlation between the neutralizing antibodies and cytokines produced after vaccination but found little to no relationship between the two (Appendix A). For different cytokines, we also drew an ROC curve to determine whether or not the number of weeks could be used to classify cytokine concentration based on the AUC value. An AUC of at least 0.8 indicates that it can be used as a good evaluation criterion. The comparative graphs and results are provided in Figure 8 and Table 6.

## 4. Discussion

Maternal immunization is a universally recognized concept, believed to play a pivotal role in reducing the susceptibility of newborns to pathogenic infections. The post-vaccination immune response, however, is influenced by a myriad of factors, including age, gender, and immune status [35,36,37,38]. Pregnancy, a unique physiological state, leads to notable shifts in steroid hormone levels, impacting immune cell functionality [39,40,41]. While there have been studies comparing the vaccine responses between pregnant and non-pregnant individuals [42,43], there remains a paucity of research on such responses in regard to SARS-CoV-2 vaccination.

Previous research indicates that microRNA is instrumental in modulating the signaling pathways of the immune system [44,45,46]. However, the role of microRNA in post-vaccination immune modulation, particularly in regard to the effects of mRNA vaccines on inflammatory cytokine profiles, warrants further investigation. Our study endeavors to elucidate the effects of COVID-19 vaccine and vaccine regimens on the immune regulatory system of pregnant women by examining microRNA expression patterns. To compile a comprehensive microRNA expression profile post-COVID-19 vaccination, we selected samples from 3 vaccinated non-pregnant individuals and 13 pregnant individuals. After excluding four outliers, we analyzed a total of 12 samples. Our findings reveal an upregulation in the expression of eight microRNAs. Of these, three microRNAs, namely miR-451a, miR-23a-3p, and miR-21-5p, displayed the most pronounced expression changes. We thus chose these three candidate microRNAs for the in-depth regulation function analysis to further delve into the immune response post-vaccination.

Our pilot study found that all the pregnant women who received the mRNA COVID-19 vaccine experienced only mild local reactions at the injection site, such as pain, redness, or swelling. Notably, no participants reported systemic side effects that required medical attention or reporting to the adverse event monitoring system. The absence of significant side effects in our study population is crucial, as it suggests that the immune response observed in our investigation is not confounded by adverse reactions to the vaccine. This finding underscores the importance of our results, as it demonstrates that the mRNA COVID-19 vaccine is well tolerated in pregnant women and that the immune response we characterized is likely a direct result of the vaccine itself rather than a consequence of any side effects.

In our investigation, we discerned that the levels of S1RBD IgG, along with cytokines IL-6, IL-6R, TNF-α, IL-10, and IL-1β, were markedly elevated in the individuals who received two or three doses of the mRNA COVID-19 vaccine compared to those who received a single dose. This observation underscores the efficacy of booster vaccinations in inducing robust protective cytokine responses post-mRNA COVID-19 vaccination. While the concentration of S1RBD IgG wanes over time post-vaccination, we noted that the cytokine levels peaked between 5 and 6 weeks after vaccination. Interestingly, no correlation was observed between the concentrations of S1RBD IgG and cytokine levels. This distinction sheds light on the divergent roles of B cells and T cells post-vaccination. While B cells generate neutralizing antibodies that thwart viral entry into cells, T cell immunity is pivotal in halting disease progression and safeguarding against emergent viral variants.

The previous literature has emphasized the cardinal role of IL-6 in both innate and adaptive immune regulation, with IL-6 known to activate B cells into antibody-secreting plasma cells [47,48,49]. However, our study did not identify any correlation between antibody concentrations and the levels of IL-6 or IL-6R. Furthermore, cytokines such as IL-6, TNF-α, IL-10, and IL-1β have been previously linked to adverse effects post-vaccination and infections. Research on mycoplasma pneumoniae membrane lipoprotein vaccine has demonstrated that lipoprotein-induced vaccine-enhanced disease in a mouse model is concomitant with elevated inflammatory cytokines in lung lavage fluid [50]. The elevated serum levels of IL-6, IL-10, and TNF-α have been correlated with disease severity and mortality in COVID-19 patients [51,52,53]. The cytokine triad of IL-1β, IL-6, and TNF is associated with the post-acute sequelae of COVID-19 [54].

Based on our research, the mRNA COVID-19 vaccine has been found to induce the upregulation of three specific microRNAs: miR-146a-5p, miR-21-5p, and miR-142-3p. These microRNAs are associated with the production of cytokines IL-6, IL-10, and TNF-α, which are known to be generated during the natural immune response to SARS-CoV-2 virus infection in humans. This suggests that the mRNA COVID-19 vaccine stimulates innate immunity in a similar manner to a natural infection with the SARS-CoV-2 virus. Given the potential implications of these cytokines in immune response, it is imperative that further research be undertaken to comprehensively understand their significance post-vaccination. While prior research has indicated that heterologous boosting can stimulate a more potent antibody response compared to homologous boosting, our study did not observe any influence of heterologous boosting on the cytokine concentrations under investigation. Additionally, the administration of pertussis and influenza vaccines during pregnancy did not exhibit any discernible impact on cytokine levels.

The main limitation of our pilot study is the inequality in sample sizes among the vaccination groups, particularly the small number of participants in the single-dose group. Due to the high vaccination rate among the pregnant women and the rarity of individuals receiving only one dose in the current climate, it was extremely challenging to achieve balanced group sizes despite our best efforts to maintain the integrity of our research data. The smaller sample size in the single-dose group may have limited the statistical power to detect significant differences compared to the other groups. While we attempted to control for potential confounding factors by maintaining a consistent interval between the last dose of vaccination and delivery (2–8 weeks) based on our previous findings, the timing of vaccination during pregnancy varied among the participants. This variation could have influenced the observed immune responses. Future studies should aim for larger sample sizes and more balanced group distributions to further validate our findings. However, this may prove difficult given the current challenges in recruiting single-dose individuals. Researchers should also strive for tighter control over the timing of vaccination during pregnancy to minimize potential confounding effects.

The second limitation of our pilot study is the lack of a direct comparison group consisting of pregnant women who developed natural immunity after COVID-19 exposure. This prevents us from quantitatively comparing the extent of the microRNA and cytokine increases between the vaccinated individuals and those with natural immunity. Future research should aim to include a cohort of pregnant women who have recovered from COVID-19 and compare their microRNA and cytokine profiles to those of vaccinated pregnant women. This would provide valuable insights into the quantitative differences in the immune response generated by natural infection versus vaccination in this specific population.

The third study limitation is the heterogeneity in the primary vaccination series among the three-dose recipients, who received either the Moderna, Oxford/AstraZeneca, or Pfizer-BioNTech BNT162b2 vaccines for their first two doses. Although this variation could potentially influence immune response, the evidence suggests that the type of primary series may not significantly impact the effectiveness of the Moderna booster dose [55]. As all the three-dose recipients in our study received the Moderna vaccine as their final dose, we believe that the impact of the heterogeneous primary series on our findings is likely to be minimal. Furthermore, our study focused primarily on comparing the immune responses between the dosage groups rather than directly comparing the vaccine types, and by ensuring that all the participants within each dosage group received the same vaccine for their last dose, we minimized the potential confounding effects of vaccine type on our results.

Lastly, our pilot study lacks comprehensive control data from non-pregnant healthy women. While our research focused on the immunological profiles of pregnant women post-COVID-19 mRNA vaccination, comparing these findings with those of non-pregnant individuals would have provided deeper insights into any potential differences in vaccine-induced immune responses. Our study included a small control group of three non-pregnant healthy women, which was insufficient for a robust comparative analysis. Future studies should include a larger cohort of non-pregnant healthy women to enhance the statistical power and comprehensively compare the immune responses between pregnant and non-pregnant individuals. This will help clarify how pregnancy-related immunological changes might influence vaccine efficacy and immune regulation.

Our research modestly attempts to shed light on the immune response induced by mRNA COVID-19 vaccination. Through our observations, we have noted the importance of cytokines and have begun to explore the potential nuances of microRNA expression patterns. We hope our findings contribute to the broader understanding of this topic.

## 5. Conclusions

Our analysis reveals the significant impact of the mRNA COVID-19 vaccine on the immune regulatory system in pregnant women, particularly concerning specific microRNA expression patterns. Notably, miR-451a, miR-23a-3p, and miR-21-5p showed increased expression post-vaccination, suggesting their potential role in immune system signaling. We also identified changes in S1RBD IgG and cytokine (IL-6, IL-6R, TNF-α, IL-10, and IL-1β) levels in the vaccinated individuals, pointing to an augmented inflammatory response in boosting individuals. Importantly, these immune markers were not correlated with S1RBD IgG concentrations. In essence, the mRNA COVID-19 vaccine essentially influences the immune response in pregnant women, with specific microRNAs playing a pivotal role. These insights set the stage for deeper studies into the intricate relationship between vaccines, microRNAs, and the immune system, especially during pregnancy.

## Figures and Tables

**Figure 1 vaccines-12-00658-f001:**
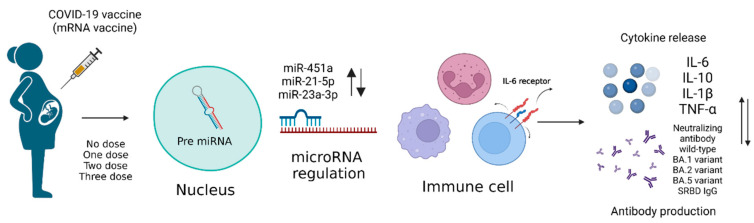
Illustrates the experimental design of our pilot study. Peripheral blood samples were collected from pregnant women who received mRNA COVID-19 vaccines, and these samples were used for microRNA analysis and ELISA. The microRNA analysis was performed to identify differentially expressed microRNAs post-vaccination, while ELISA was used to measure the concentrations of S1RBD IgG and neutralizing antibodies against wild-type and different variants, and various cytokines, including IL-6, IL-6R, TNF-α, IL-10, and IL-1β (created using BioRender.com, accessed on 9 June 2024).

**Figure 2 vaccines-12-00658-f002:**
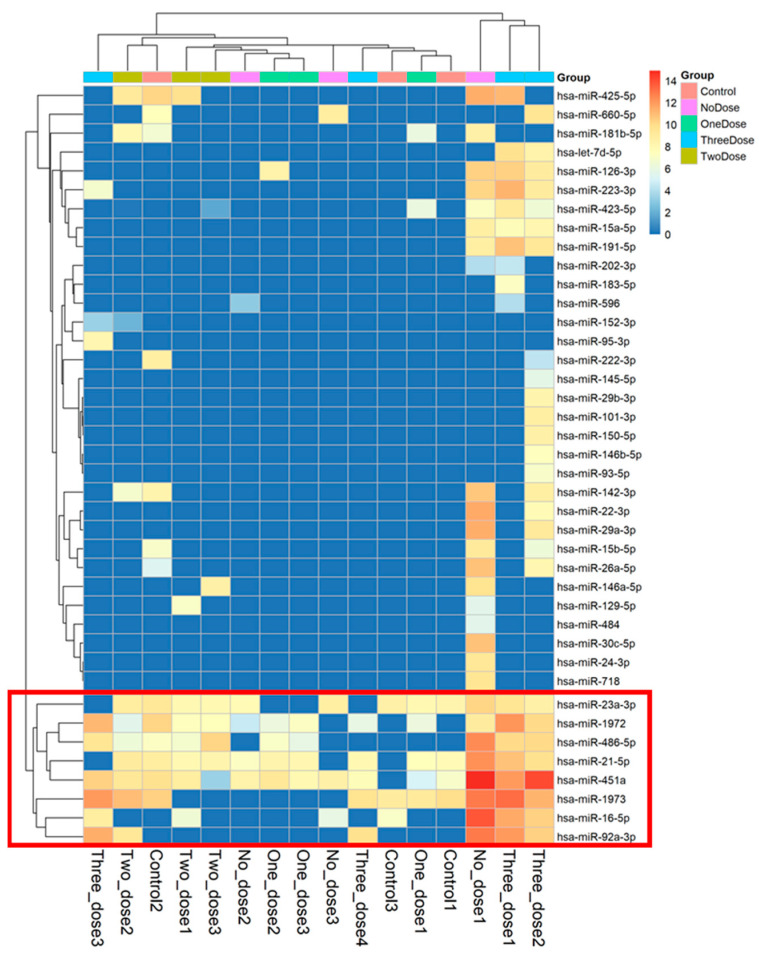
Heatmap of microRNA expression profiles from the 16 peripheral blood samples of the pregnant women who received varying doses of the mRNA COVID-19 vaccine and the non-pregnant controls. The heatmap visualizes the relative expression levels of microRNAs, with eight microRNAs (highlighted with red frame lines) exhibiting distinct expression patterns among the sample groups.

**Figure 3 vaccines-12-00658-f003:**
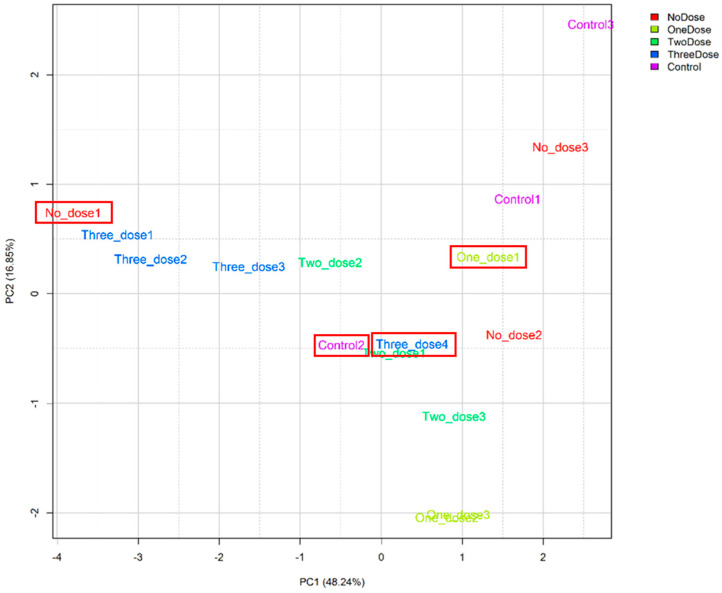
PCA graph of the 16 samples (red circles on the graph are 4 outliers). All 16 samples were divided into five groups based on the administered vaccine dosage. After performing the PCA analysis, we observed that four samples (No_dose1, One_dose1, Control2, and Three_dose4) were distant from the other samples within their respective groups, indicating potential outliers that could affect the analysis results. Therefore, these samples were excluded from subsequent analyses.

**Figure 4 vaccines-12-00658-f004:**
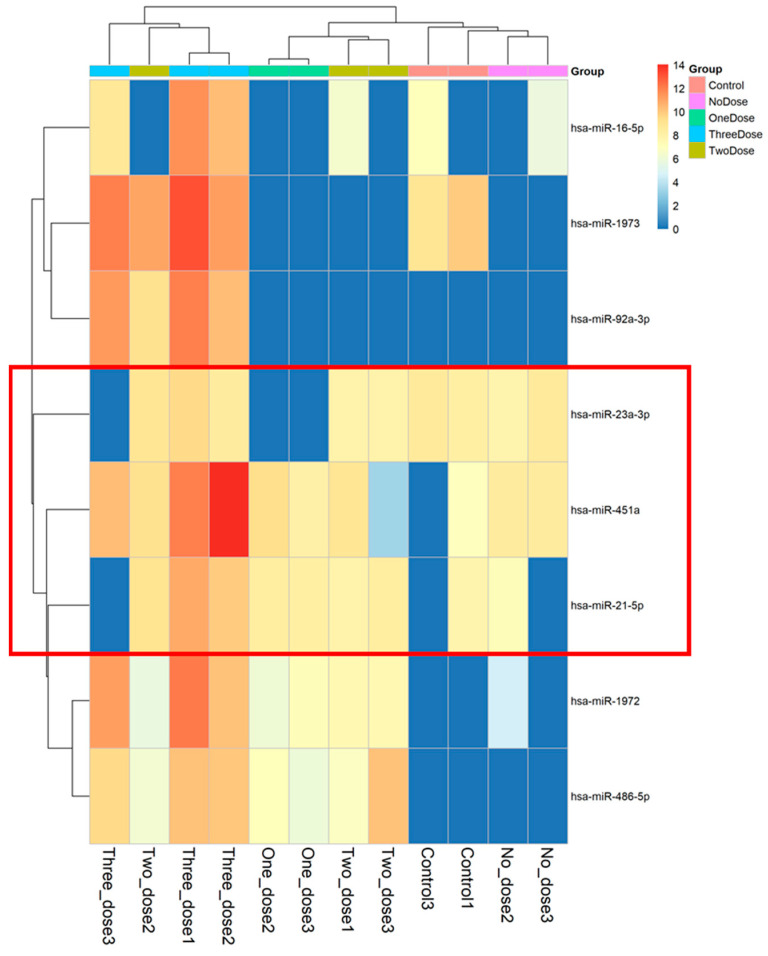
Focused heatmap of the 8 microRNAs with significant expression differences among the 12 peripheral blood samples. The three microRNAs highlighted with red frame lines (miR-451a, miR-23a-3p, and miR-21-5p) show the most pronounced expression changes and were selected for further investigation of their potential roles in the immune response to the mRNA COVID-19 vaccination during pregnancy.

**Figure 5 vaccines-12-00658-f005:**
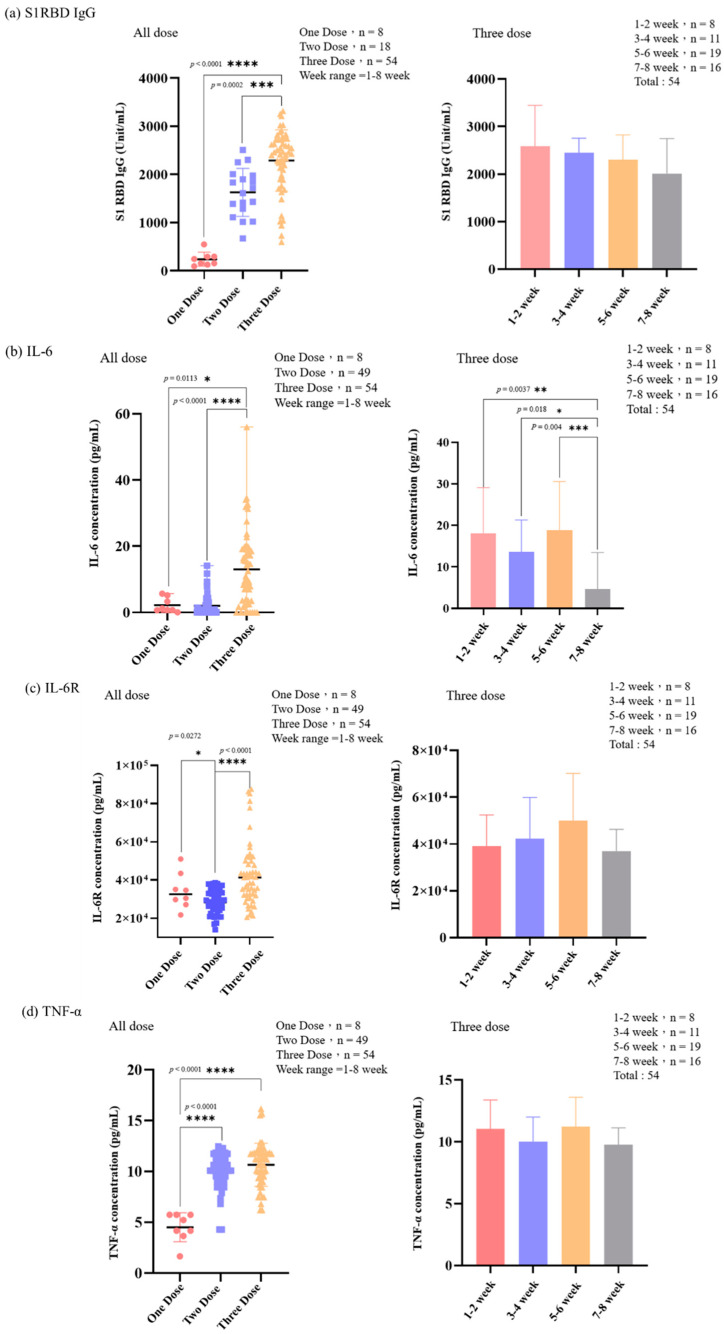
(**a**) S1RBD IgG, (**b**) IL-6, (**c**) IL-6R, (**d**) TNF-α, (**e**) IL-10, and (**f**) IL-1β concentrations. The figures on the left show the samples classified according to the number of injections. The figures on the right show the samples classified by the number of weeks between the third dose of injection time and the day of delivery. * *p* < 0.05, ** *p* < 0.01, *** *p* < 0.001, **** *p* < 0.0001.

**Figure 6 vaccines-12-00658-f006:**
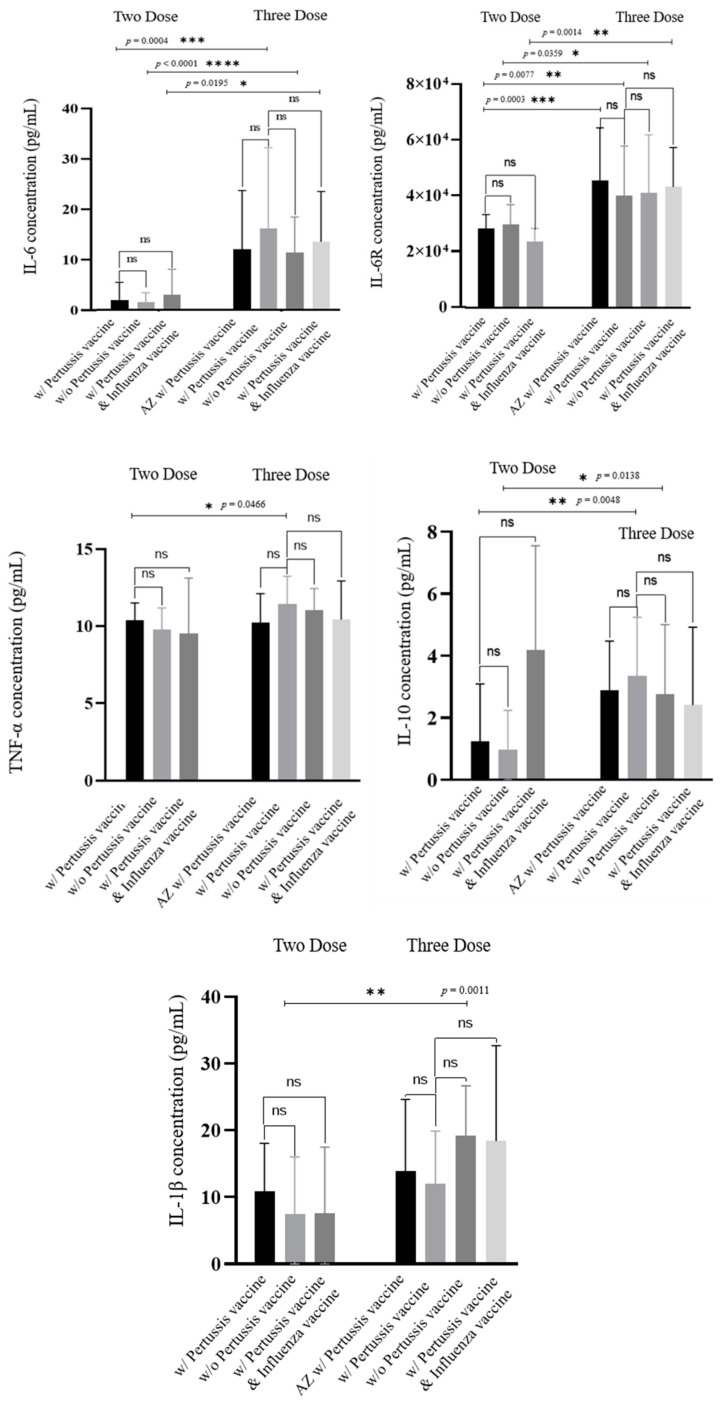
IL-6, IL-6R, TNF-α, IL-10, and IL-1β concentrations (week range = 1–8 weeks; two doses with pertussis vaccine, n = 21, without pertussis vaccine, n = 21, and with pertussis and influenza vaccines, n = 7; three doses with AZ and pertussis vaccines, n = 13, only with pertussis vaccine, n = 11, without pertussis vaccine, n = 7, and with pertussis and influenza vaccines, n = 23. Correlation of all cytokines and S1RBD IgG. ns, *p* > 0.05, * *p* < 0.05, ** *p* < 0.01, *** *p* < 0.001, **** *p* < 0.0001.

**Figure 7 vaccines-12-00658-f007:**
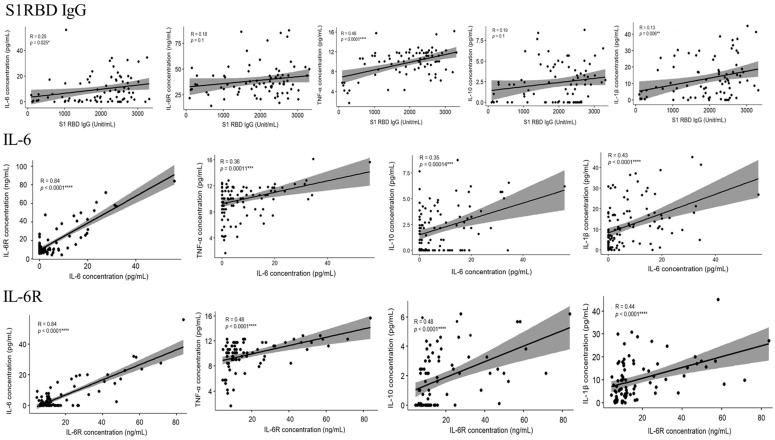
The correlation analysis between the five cytokines (IL-6, IL-6R, TNF-α, IL-10, and IL-1β) and S1RBD IgG. The analysis was performed using the R software to determine whether there were any significant relationships among these immune markers. The Spearman’s rank correlation coefficients (R) and probability density values (*p*) are provided for each pairwise comparison. ** *p* < 0.01, *** *p* < 0.001, **** *p* < 0.0001.

**Figure 8 vaccines-12-00658-f008:**
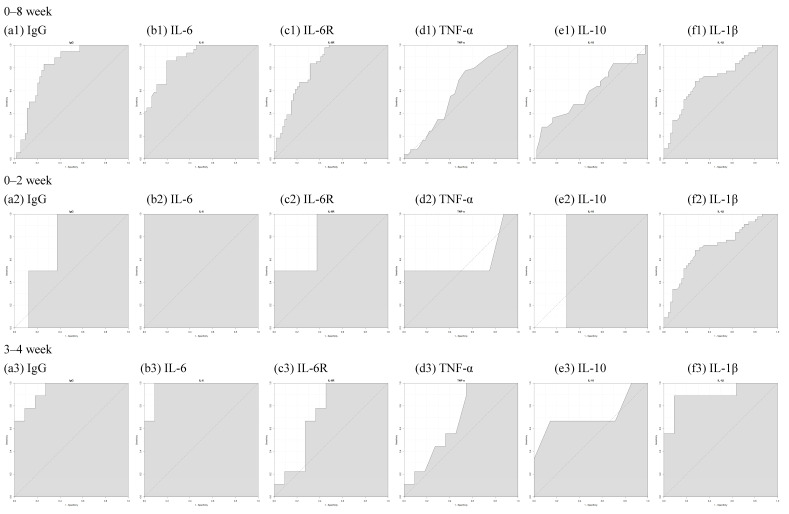
The receiver operating characteristic (ROC) curves for S1RBD IgG and the five examined cytokines (IL-6, IL-6R, TNF-α, IL-10, and IL-1β) over different time intervals post-vaccination. The ROC curves were generated to evaluate the ability of these immune markers to discriminate between different time points after vaccination. The area under the curve (AUC) values were calculated for each marker and time interval, with an AUC of at least 0.8 indicating good discriminatory power.

**Table 1 vaccines-12-00658-t001:** Participant demographic and clinical data for microRNA analysis.

	Clinical Data					
Sample ID	Parity	Age (Year)	BMI	Weeks of Gestation at Delivery (Weeks)	The Interval between the Last Dose of the COVID-19 Vaccine and the Day of Delivery (Weeks)	Kind of COVID-19 Vaccine for the First/Second/Third Dose
Control 1	-	23	19.8	-	-	BNT/BNT/Moderna
Control 2	-	23	18.62	-	-	BNT/BNT/Moderna
Control 3	-	23	21	-	-	BNT/BNT/Moderna
No dose 1	3	40	28.38	39	-	-
No dose 2	1	40	26.95	37	-	-
No dose 3	4	38	30.12	36	-	-
One dose 1	1	36	25	36	15	Moderna/-/-
One dose 2	2	25	26.8	37	6	Moderna/-/-
One dose 3	0	37	25	39	6	Moderna/-/-
Two dose 1	3	40	25.7	39	8	Moderna/Moderna/-
Two dose 2	0	31	25.71	38	5	Moderna/Moderna/-
Two dose 3	1	33	28	40	4	Moderna/Moderna/-
Three dose 1	2	35	21.8	40	6	Moderna/Moderna/Moderna
Three dose 2	1	24	27.6	38	5	Moderna/Moderna/Moderna
Three dose 3	1	32	23.82	40	5	Moderna/Moderna/Moderna
Three dose 4	2	35	23.52	39	6	Moderna/Moderna/Moderna

**Table 2 vaccines-12-00658-t002:** Participant demographic and clinical data for ELISA.

Variable	Included in the Analysis		
	One Dose ^a^	Two Dose ^b^	Three Dose ^c^
Age of mothers (years)	34.25 * ± 6.58 ** IQR 38–28.5	32.49 ± 4.66 IQR 35–28	33.57 ± 5.36 IQR 37.75–30
Parity	1.375 * ± 0.74 ** IQR 1.25–0	1.31 ± 0.55 IQR 2–1	1.57 ± 0.63 IQR 2–1
BMI (body mass index)	26.93 * ± 4.13 ** IQR 28.19–23.68	27 ± 4.41 IQR 30.3–23.82	27.29 ± 3.99 IQR 29.61–24.3
Weeks of gestation at the first dose of COVID-19 vaccination (weeks)	33.5 ± 2.27 IQR 35–32.5	23.06 ± 10.26 IQR 28–24	0.74 ± 8.93 IQR 4.75–(−5.57)
Weeks of gestation at the second dose of COVID-19 vaccination (weeks)	-	32.82 ± 1.91 IQR 34–31	14.48 ± 7.29 IQR 20–9.5
Weeks of gestation at the third dose of COVID-19 vaccination (weeks)	-	-	33.33 ± 2.49 IQR 35–31
The interval between the first dose of COVID-19 vaccination and the collection of blood samples (the day of delivery) (weeks)	4.75 ± 1.91 IQR 6–3.75	15.27 ± 10.08 IQR 15–10	37.43 ± 9.05 IQR 42–34
The interval between the second dose of COVID-19 vaccination and the collection of blood samples (the day of delivery) (weeks)	-	5.67 ± 1.76 IQR 7–5	23.72 ± 7.39 IQR 29–19
The interval between the third dose of COVID-19 vaccination and the collection of blood samples (day of delivery) (weeks)	-	-	5.11 ± 2.03 IQR 7–4
Weeks of gestation at delivery (weeks)	38.25 ± 0.89 IQR 39–37.75	38.49 ± 1.17 IQR 39–38	38.44 ± 1.11 IQR 39–38
Weight of newborn (g)	3123.13 * ± 363.52 ** IQR 3205–2992.5	3089.9 ± 360.78 IQR 3235–2900	3103.89 ± 311.99 IQR 3293.75–2883.8
Sex of newborn			
Male	4 (50% ***)	22 (44.9% ***)	28 (51.9% ***)
Female	4 (50% ***)	27 (55.1% ***)	26 (48.1% ***)

* mean; ** standard deviation (± SD); *** percentage of all the surveyed patients. ^a^ = one dose, n = 8; ^b^ = two dose, n = 49; ^c^ = three dose, n = 54; IQR: interquartile range.

**Table 3 vaccines-12-00658-t003:** Top 5 enriched terms for miR-451a.

miR-451a		
Pathway Description	Gene Name	*p*-Value
Interleukin-4 and Interleuki-13 Signaling	AKT1/MMP2/MMP9/BCL2/ MYC/IL6R	2.60 × 10^−7^
Vascular Inflammations	AKT1/MMP2/MMP9/ADAM10/CRP/ ETS1/CAV1	4.09 × 10^−7^
Immune Thrombocytopenic Purpura	MIF/ABCB1/MMP9/BCL2/IKBKB/ ADAM10/CRP	9.75 × 10^−7^
Acquired Immunodeficiency Syndrome	ABCB1/MMP2/MMP9/BCL2/ MYC/CRP	2.31 × 10^−6^
Pelvic Inflammatory Disease	MMP2/MMP9/CRP	1.32 × 10^−5^
Target Cytokine related to COVID-19 [23,24,25]	IL-6R/IL-6/TNF-α [23,24,25]	

**Table 4 vaccines-12-00658-t004:** Top 5 enriched terms for miR-21-5p.

miR-21-5p		
Pathway Description	Gene Name	*p*-Value
Inflammation	PTEN/FAS/EGFR/IL1B/ICAM1/PLAT/PTX3/TNFAIP3/CCL20/DUSP10/PPARA/NTF3/FASLG/SMAD7/MMP2/VEGFA/MMP9/STAT3/MYD88/IL12A/CXCL10/CLU/CCR7/TLR2	2.79 × 10^−11^
Interleukin-4 and Interleukin-13 Signaling	BCL2/SOCS5/MYC/IL1B/ICAM1/FASLG/SOX2/MMP2/VEGFA/MMP9/STAT3/BCL6/IL12A/FOXO3	6.62 × 10^−10^
Acquired Immunodeficiency Syndrome	BCL2/PTEN/FAS/MYC/IL1B/ICAM1/SP1/FASLG/MMP2/VEGFA/MMP9/SLK/STAT3/BCL6/MYD88/CXCL10	1.96 × 10^−9^
Immune Thrombocytopenic Purpura	RASGRP1/BCL2/JAG1/TIMP3/PTEN/FAS/IL1B/TNFAIP3/FASLG/SMAD7/VEGFA/MMP9/STAT3/BCL6/IL12A/CXCL10/GP5	3.41 × 10^−8^
Vascular Inflammations	TIMP3/PTEN/PPIF/IL1B/ICAM1/PTX3/SMARCA4/SMAD7/MMP2/VEGFA/MMP9/AKT2/STAT3/FOXO3/CXCL10/TLR2	4.90 × 10^−8^
Target Cytokine related to COVID-19 [26,27]	IL-1β/IL-12 [26,27]	

**Table 5 vaccines-12-00658-t005:** Top 5 enriched terms for miR-23a-3p.

miR-23a-3p		
Pathway Description	Gene Name	*p*-Value
Inflammation	CXCL12/PTEN/CDH1/IRF1/FAS/MT2A/STS/VCAM1/TNFAIP3/TLR6/FGF2/FOXP3/SIRT1	2.30 × 10^−5^
Human Immunodeficiency Virus Infectious Disease	CXCL12/CDH1/IRF1/FAS/VCAM1/SOD2/ MEF2C/FOXP3/BCL2/SIRT1	2.96 × 10^−5^
Experimental Autoimmune Encephalomyelitis	FOXO3/VCAM1/ADAM17/FGF2/ TGFB2/SIRT1	5.85 × 10^−5^
Autoimmune Lymphoproliferative Syndrome	FAS/TNFAIP3/XIAP/FOXP3/BCL2	8.79 × 10^−5^
Immune Thrombocytopenic Purpura	CXCL12/HES1/PTEN/FAS/HOXB4/STS/ TNFAIP3/FGF2/FOXP3/BCL2	0.000127
Target Cytokine or Protein related to COVID-19 [24]	IL-6/BCL2 [24]	

**Table 6 vaccines-12-00658-t006:** AUC of S1RBD IgG, IL-6, IL-6R, TNF-α, IL-10, and IL-1β over time.

	0–8 Week	0–2 Week	3–4 Week	5–6 Week	7–8 Week
S1RBD IgG	0.8086	0.75	0.9394	0.9158	0.6875
IL-6	0.8887	1	0.9697	0.9323	0.6302
IL-6R	0.8107	0.8125	0.7273	0.8421	0.898
TNF-α	0.5911	0.5938	0.6717	0.6648	0.5362
IL-10	0.5865	0.7143	0.7143	0.6545	0.5357
IL-1β	0.7146	0.8571	0.899	0.7288	0.6039

## Data Availability

The data presented in this study are available in this article and Appendix A.

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
