# Peer review of "COVID-19 Vaccination in Pregnancy: Pilot Study of Plasma MicroRNAs Associated with Inflammatory Cytokines after COVID-19 mRNA Vaccination"

_vaccines, 2024, doi:10.3390/vaccines12060658_

Round 1

Reviewer 1 Report

Comments and Suggestions for Authors

The manuscript submitted to vaccines by Shen et al., titled: "COVID-19 Vaccination in Pregnancy: Plasma MicroRNAs Associated with Inflammatory Cytokines after COVID-19 mRNA Vaccination" is an interesting human trial aiming to investigate the effect of mRNA vaccination agains COVID-19 in pregnant women in terms of inflammatory cytokine production. 

The work is important in terms of informing clinical practice but also preventive public health initiatives. The work is well written and structured logically. The reviewer would like to raise the following points for the authors' consideration:

1. What were the inclusion and exclusion criteria for the participants?

2. Was the stage of pregnancy considered?

3. How was the number of participants determined?

4. Was the type of mRNA vaccine considered?

5. The authors saw an increase in specific mRNAs that involve the production of cytokines that are indicative of the inflammatory / immune response that is seen how does this increase compare (not only qualitatively but also quantitatively) with the one seen through natural immunity developed secondary to COVID-19 exposure?

6. Is the persistence (half life) of mRNAs the same in vaccine versus natural immunity?

7. Have side-effects been considered in the discussion?

Author Response

The manuscript submitted to vaccines by Shen et al., titled: "COVID-19 Vaccination in Pregnancy: Plasma MicroRNAs Associated with Inflammatory Cytokines after COVID-19 mRNA Vaccination" is an interesting human trial aiming to investigate the effect of mRNA vaccination agains COVID-19 in pregnant women in terms of inflammatory cytokine production. The work is important in terms of informing clinical practice but also preventive public health initiatives. The work is well written and structured logically. The reviewer would like to raise the following points for the authors' consideration:

  1. What were the inclusion and exclusion criteria for the participants?

Response: Thank you for this valuable comment. We have revised our manuscript to include a detailed description of the inclusion and exclusion criteria, as highlighted in yellow (line 97-103, Page 3).

  1. Was the stage of pregnancy considered?

Response: Thank you for this valuable comment. In our study, we specifically selected cases where the interval between the last dose of the vaccine and delivery was between 4-8 weeks. This decision was based on our previous research and findings from other research groups, which have shown that the antibody concentration generated during this period was at its peak (plateau phase). This time window ensured the highest antibody transfer efficiency to the fetus as well. We have revised our manuscript, as highlighted in yellow (line 103-106, Page 3).

  1. How was the number of participants determined?

Response: Thank you for this comment. The sample size of our study was determined based on a power analysis, considering the desired effect size, power, and confidence level. We aimed to have a medium effect size (Cohen's d = 0.5) with a power of 80% and a 95% confidence level (α = 0.05). These parameters were chosen per conventional standards in the field to ensure that our study had sufficient statistical power to detect meaningful differences between the groups.

  1. Was the type of mRNA vaccine considered?

Response: Thank you for this comment. As we mentioned in our revised manuscript (line 112-116, Page 3), “Both the two-dose vaccine and the one-dose vaccine recipients received Moderna (mRNA-1273) vaccine. Among the three-dose recipients, all received the Moderna vaccine as their third dose, whereas their first and second doses were sets of either the Moderna, the Oxford/AstraZeneca, or Pfizer-BioNTech BNT162b2 vaccine.”

A study published in New England Journal of Medicine, COVID-19 Vaccine Effectiveness against the Omicron (B.1.1.529) Variant (N Engl J Med, 2022, 386, pp. 1532-1546), found that regardless of whether the primary series consisted of the ChAdOx1 nCoV-19 or BNT162b2 vaccine, the vaccine effectiveness after receiving the Moderna booster dose was similar. Based on this evidence, we believe that the variation in the primary series among our three-dose recipients was unlikely to have a significant impact on our findings, as all participants in this group received the Moderna vaccine as their booster dose. Furthermore, our study primarily focused on comparing the immune responses between different dosage groups (one, two, or three doses) rather than directly comparing the effects of different vaccine types. By ensuring that all participants within each dosage group received the same type of vaccine (Moderna) for their last dose, we minimized the potential confounding effects of vaccine type on our results. We have included this information in our revised manuscript to clarify our consideration of vaccine types and to address any concerns regarding the potential impact of heterogeneous primary series on our findings, as highlighted in yellow (line 415-425, Page 20 & 21).

  1. The authors saw an increase in specific mRNAs that involve the production of cytokines that are indicative of the inflammatory/immune response that is seen how does this increase compare (not only qualitatively but also quantitatively) with the one seen through natural immunity developed secondary to COVID-19 exposure?

Response: Thank you for this valuable comment. In our study, we observed an increase in specific microRNAs (miR-451a, miR-23a-3p, and miR-21-5p) that are associated with the production of inflammatory cytokines (IL-6, IL-10, and TNF-α) following mRNA COVID-19 vaccination. These cytokines are known to be generated during the natural immune response to SARS-CoV-2 infection. However, due to the lack of a direct comparison group of pregnant women who developed natural immunity after COVID-19 exposure, we were unable to quantitatively compare the extent of the microRNA and cytokine increases between the two scenarios. Qualitatively, our findings suggest similarities between the immune responses induced by mRNA COVID-19 vaccines and natural SARS-CoV-2 infection, as evidenced by the upregulation of specific microRNAs and associated inflammatory cytokines. To quantitatively compare the magnitude of these changes, future research should include a cohort of pregnant women who have recovered from COVID-19 and compare their microRNA and cytokine profiles to those of vaccinated pregnant women. We have addressed this limitation in our revised manuscript, acknowledging that the lack of a direct comparison group prevents us from making quantitative comparisons between the immune responses induced by vaccination and natural infection in pregnant women, as highlighted in yellow (line 406-414, Page 20). We appreciate you raising this important point, as it highlights the need for further research to better understand the differences and similarities between these two scenarios.

  1. Is the persistence (half life) of mRNAs the same in vaccine versus natural immunity?

Response: Thank you for this comment. There is currently a lack of direct comparative studies regarding whether microRNAs' persistence (half-life) is the same in vaccine-induced immunity versus natural immunity for COVID-19. Our study design was carefully crafted to limit the time between the last dose of the vaccine and delivery to 4-8 weeks. This decision was made to minimize the potential confounding effects of time on the observed microRNA and cytokine levels. By focusing on a specific time window, we aimed to capture a more stable representation of the immune response post-vaccination, reducing the impact of time-dependent variations in data. Future research could longitudinally follow pregnant women who received mRNA COVID-19 vaccines and those who recovered from COVID-19, comparing the changes in microRNA expression levels over time to understand if their persistence and half-life are similar.

  1. Have side-effects been considered in the discussion?

Response: Thank you for raising the important question about side effects following mRNA COVID-19 vaccination in pregnant women. In our study, all the women who received the vaccine reported only mild local reactions at the injection site, such as pain, redness, or swelling. None of the participants experienced systemic side effects that required medical attention or reporting to the adverse event monitoring system. We acknowledge that discussing side effects is an essential aspect of any study investigating the impact of vaccination. In response to your comment, we have revised the manuscript to include a brief statement addressing the absence of significant side effects in our study population, as highlighted in yellow (line 346-355, Page 19).

Reviewer 2 Report

Comments and Suggestions for Authors

Authors explored immunological profile changes triggered by the mRNA vaccines of COVID-19 in 111 pregnant women. Expression of miRNAs were analyzed after vaccination and it was found that three miRNAs (miR-451a, has-miR-23a-3p, and has-miR-21-5p) were significantly upregulated. These miRNAs are associated with the production of cytokines IL-6, IL-10, and TNF-α, which are known to be generated during the natural immune response to SARS-CoV-2 virus infection in humans. On the other hand, no correlation was observed between the concentrations of S1RBD IgG and cytokine levels, including Il-6 and IL-6R. Although this study sheds some light on the immune response induced by mRNA COVID-19 vaccination, the intricate interplay between vaccines, miRNAs, and cytokines and immune responses must be explored further. The manuscript also leaves ample room for improvement with regard to data interpretation and detailed explanations.

Major points:

The supplementary figures S1 and S2 were not provided.

The figures must be explained in detail in results section. The same applies to figures legends, if possible.

3. line 46: As of now in 2022 -> must be updated.

4. Legend to Figures 1, 7 and 8 must have explanations for the figures, not just the title only.

5. In Figures 2 and 4, red frame lines are not shown. As with Figure 1, the legends must explain the figures.

Minor points:

1. In Table 1, IQR must be explained.

2. In Tables 3, 4, and 5, the last row (Target cytokine or proteinrelated to COVID-19) must be explained, including how the genes were selected.

3. The graphs in Figure 6, 7 and 8 are difficult to read. The fonts may well be enlarged.

Comments on the Quality of English Language

The manuscript must be edited by a native speaker of English for clarity and usages.

Author Response

111 pregnant women. Expression of miRNAs were analyzed after vaccination and it was found that three miRNAs (miR-451a, has-miR-23a-3p, and has-miR-21-5p) were significantly upregulated. These miRNAs are associated with the production of cytokines IL-6, IL-10, and TNF-α, which are known to be generated during the natural immune response to SARS-CoV-2 virus infection in humans. On the other hand, no correlation was observed between the concentrations of S1RBD IgG and cytokine levels, including Il-6 and IL-6R. Although this study sheds some light on the immune response induced by mRNA COVID-19 vaccination, the intricate interplay between vaccines, miRNAs, and cytokines and immune responses must be explored further. The manuscript also leaves ample room for improvement with regard to data interpretation and detailed explanations.

Major points:

The supplementary figures S1 and S2 were not provided.

Response: Thank you for this comment. We have attached Supplementary Figures S1 and S2.

The figures must be explained in detail in results section. The same applies to figures legends, if possible.

Response: Thank you for this comment. We have significantly revised our manuscript.

line 46: As of now in 2022 -> must be updated.

Response: Thank you for this comment. We have updated the year, as highlighted in yellow (line 45, Page 2).

Legend to Figures 1, 7 and 8 must have explanations for the figures, not just the title only.

Response: Thank you for this comment. We have revised the figure captions of Figures 1, 7 and 8 in our revised manuscript, as highlighted in yellow.

In Figures 2 and 4, red frame lines are not shown. As with Figure 1, the legends must explain the figures.

Response: Thank you for this comment. We have revised the figure caption of Figure 1, as highlighted in yellow, and re-prepared Figures 2 and 4 in our revised manuscript.

Minor points:

In Table 1, IQR must be explained.

Response: Thank you for this comment. We have explained IQR in our revised manuscript, as highlighted in yellow (line 196, Page 6).

In Tables 3, 4, and 5, the last row (Target cytokine or protein related to COVID-19) must be explained, including how the genes were selected.

Response: Thank you for this comment. We selected these cytokines/proteins based on 1) the gene name (related to the pathway description) that we listed in Tables 3, 4 and 5, and 2) the previous lectures that we mentioned in our revised manuscript (line 78-84, Page 2), “There are also many immune-related cytokines associated with the regulation of microRNAs. For example, miR-451a is an IL-6R translational repressor that exacerbates the IL-6-induced cytokine storm23 and one of the top 5 downregulated microRNA genes in COVID-19 patients24. One study demonstrated that the presence of miR-451a significantly suppressed TNF-α and significantly increased IL-1025. Another scientific study showed that miR-21-5p had an anti-inflammatory effect that could inhibit the expression of IL-1β26, 27.”

We have added the relevant references in Tables 3, 4 and 5.

The graphs in Figure 6, 7 and 8 are difficult to read. The fonts may well be enlarged.

Response: Thank you for this comment. We have fixed this issue.

Reviewer 3 Report

Comments and Suggestions for Authors

the article offers a good pharmacological explanation of covid-19 vaccines for pregnant women and a fine scenario of pregnancy related outcomes (I.e. abortion , low birth weight newborns, pre term delivery).

yet, common covid-19 vaccines side effects or their pathological phenotypes are completely absent: venous thrombosis , based also on the fact that pregnancy as covid 19 vaccines is an environmental thrombotic risk factor , and unusual/ atypical autoimmune diseases . Laboratory biomarkers to monitor these types of outcomes are lacking too.

Author Response

The article offers a good pharmacological explanation of covid-19 vaccines for pregnant women and a fine scenario of pregnancy related outcomes (i.e. abortion, low birth weight newborns, pre term delivery).

Yet, common covid-19 vaccines side effects or their pathological phenotypes are completely absent: venous thrombosis, based also on the fact that pregnancy as covid-19 vaccines is an environmental thrombotic risk factor , and unusual/atypical autoimmune diseases. Laboratory biomarkers to monitor these types of outcomes are lacking too.

Response: This comment is quite out of the main goal of our study described in our manuscript. We do not know how to respond this comment and revise our manuscript based on this comment.

Reviewer 4 Report

Comments and Suggestions for Authors

  In this study, the authors attempted to elucidate the effects of COVID-19 mRNA vaccine on the immune responses of pregnant individuals. They firstly found that three miRNAs, miR-451a, has-miR-23a-3p, and has-miR-21-5p, displayed the most pronounced expression in pregnant individuals. Next, they measured the quantification of S1RBD IgG, IL-6, IL-6R, TNF-alpha, IL-10, and IL-1beta in pregnant women classified by the number of booster vaccinations or the number of weeks between vaccine administration and the time of delivery. Further, it was shown that the administration of other different vaccines did not seem to have a significant impact on the immune responses induced by COVID-19 vaccine. The idea of the experiment is unique and the data contain useful information. However, the major concern the reviewer has is that there are no data of control samples from non-pregnant healthy women in the ELISA data (Figs. 5 & 6). I have raised several points which need to be clarified. These are given below.

Specific comments:

1.     In the data for miRNA analysis, the sample size is too small (only three or four participants per each).

2.     I want to know why the authors selected only the data of 12 samples in Fig. 4. There are 16 samples in Figs. 2 & 3. How about the data of the remaining 4 samples?

3.     In Figs. 5 & 6, there are no data of control samples from non-pregnant healthy women. It would be more interesting to find out whether there is any or no difference in the vaccine-induced immune responses between pregnant and non-pregnant individuals.

Author Response

In this study, the authors attempted to elucidate the effects of COVID-19 mRNA vaccine on the immune responses of pregnant individuals. They firstly found that three miRNAs, miR-451a, has-miR-23a-3p, and has-miR-21-5p, displayed the most pronounced expression in pregnant individuals. Next, they measured the quantification of S1RBD IgG, IL-6, IL-6R, TNF-alpha, IL-10, and IL-1beta in pregnant women classified by the number of booster vaccinations or the number of weeks between vaccine administration and the time of delivery. Further, it was shown that the administration of other different vaccines did not seem to have a significant impact on the immune responses induced by COVID-19 vaccine. The idea of the experiment is unique and the data contain useful information. However, the major concern the reviewer has is that there are no data of control samples from non-pregnant healthy women in the ELISA data (Figs. 5 & 6). I have raised several points which need to be clarified. These are given below.

Specific comments:

In the data for miRNA analysis, the sample size is too small (only three or four participants per each).

Response: Thank you for this valuable comment. In this study, the microRNA analysis was indeed limited in number. We have revised the title of our revised manuscript as highlighted in yellow, i.e., a pilot study. We have also addressed this issue as the main limitation of our study in our revised manuscript, as highlighted in yellow (line 391-405, Page 20).

I want to know why the authors selected only the data of 12 samples in Fig. 4. There are 16 samples in Figs. 2 & 3. How about the data of the remaining 4 samples?

Response: Thank you for this comment. All 16 samples were divided into five groups based on the administered vaccine dosage. After performing PCA analysis, we observed that four samples (No_dose1, One_dose1, Control2, Three_dose4) were distant from the other samples within their respective groups, indicating potential outliers that could affect the analysis results. Therefore, these samples were excluded from subsequent analyses. Comparing Figure 2 (before exclusion) with Figure 4 (after exclusion), we noticed an improvement in the classification effectiveness of each group. This improvement is attributed not only to the selection of microRNAs but also to the exclusion of outlier samples. We have revised the figure caption of Figure 3, as highlighted in yellow.

In Figs. 5 & 6, there are no data of control samples from non-pregnant healthy women. It would be more interesting to find out whether there is any or no difference in the vaccine-induced immune responses between pregnant and non-pregnant individuals.

Response: Thank you for your insightful feedback. Our study aimed to investigate the specific immune responses, particularly the expression of microRNAs and cytokines, in pregnant women following COVID-19 mRNA vaccination. This demographic is of particular interest due to the unique immunological adjustments that occur during pregnancy. We agree that a comparative analysis with non-pregnant healthy women would provide deeper insights into whether pregnancy alters the vaccine-induced immune response. We have addressed your concern regarding the lack of data from non-pregnant healthy women by including it as a limitation in our revised manuscript, as highlighted in yellow (line 426-435, Page 21). Our future studies will incorporate a more robust control group of non-pregnant healthy women, allowing for a detailed comparative analysis that will provide valuable insights into the differential vaccine-induced immune responses in pregnant versus non-pregnant individuals.

Round 2

Reviewer 1 Report

Comments and Suggestions for Authors

The authors have made a reasonable effort to address reviewer's points.

Author Response

Thank you so much!

Reviewer 3 Report

Comments and Suggestions for Authors

Thrombosis has been the main discussed topic regarding the safety of all anti sars cov2 vaccines and the topic is not discussed.

autoimmune diseases have been reported after mRNA anti sars cov2 vaccine but biomarkers regarding autoimmune diseases are not reported 

Author Response

Responses to Reviewer 3 comments

Thrombosis has been the main discussed topic regarding the safety of all anti sars cov2 vaccines and the topic is not discussed. Autoimmune diseases have been reported after mRNA anti sars cov2 vaccine but biomarkers regarding autoimmune diseases are not reported.

Response: Thank you for your valuable comments about the potential side effects and pathological phenotypes associated with COVID-19 vaccine in pregnant women. Our study primarily focused on the immunological aspects of COVID-19 mRNA vaccination during pregnancy, specifically investigating the changes in microRNA expression and cytokine level. While we acknowledge the importance of monitoring potential side effects such as venous thrombosis and unusual/atypical autoimmune diseases, our study design did not include a comprehensive assessment of these outcomes. However, we would like to clarify that our study population was carefully selected based on strict inclusion and exclusion criteria. We excluded pregnant women who experienced systemic adverse reactions after vaccination, as well as those with pre-existing systemic diseases such as autoimmune disorders, cardiovascular diseases, thyroid dysfunction, and pregnancy-related complications. This approach allowed us to minimize the potential confounding effects of these factors on our immunological findings. In response to your comments, we have revised the “Participants and sample collection” in Materials & Methods to include information about the exclusion criteria related to underlying diseases, as highlighted in yellow (line 98-102, Page 3).

Regarding laboratory biomarkers, we collected and analyzed liver and kidney function tests and coagulation profile for participants who underwent cesarean section. The coagulation function test was performed for participants who received painless labor during vaginal delivery. These biochemical parameters were within normal ranges, meaning no significant adverse effects on liver, kidney, or coagulation function in our study population. We have revised our manuscript, as highlighted in yellow (line 106-111, Page 3).

Once again, thank you for your valuable comments. Your comments have pointed out the potential directions for future research, and we will consider incorporating these aspects in our ongoing work regarding COVID-19 vaccination during pregnancy.

Reviewer 4 Report

Comments and Suggestions for Authors

I have no serious criticisms in the current version of the manuscript.

Author Response

Thank you so much!

Round 3

Reviewer 3 Report

Comments and Suggestions for Authors

Dear authors, I can understand your point of view and you answers but pathophysiology of VITT or VIPIT (I.e. vaccine induced thrombosis) is mainly on autoimmune mechanisms. For this reason, I think that you draft is lacking of this link.

i’m sorry to reject it from my side 

Author Response

Responses to Reviewer 3 comment

Dear authors, I can understand your point of view and you answers but pathophysiology of VITT or VIPIT (I.e. vaccine induced thrombosis) is mainly on autoimmune mechanisms. For this reason, I think that you draft is lacking of this link.

Response: Thank you for your valuable comment. Our study primarily focused on the immunological aspects of COVID-19 mRNA vaccination during pregnancy, specifically investigating the changes in microRNA expression and cytokine level. While we acknowledge the importance of monitoring potential side effects such as venous thrombosis and unusual/atypical autoimmune diseases, our study design did not include a comprehensive assessment of these outcomes. However, we would like to clarify that our study population was carefully selected based on strict inclusion and exclusion criteria. We excluded pregnant women who experienced systemic adverse reactions after vaccination, as well as those with pre-existing systemic diseases such as autoimmune disorders, cardiovascular diseases, thyroid dysfunction, and pregnancy-related complications. This approach allowed us to minimize the potential confounding effects of these factors on our immunological findings. In response to your comments, we have revised the “Participants and sample collection” in Materials & Methods to include information about the exclusion criteria related to underlying diseases, as highlighted in yellow (line 98-102, Page 3).